# Molecular Epidemiology of Antimicrobial Resistance and Virulence Profiles of *Escherichia coli*, *Salmonella* spp., and *Vibrio* spp. Isolated from Coastal Seawater for Aquaculture

**DOI:** 10.3390/antibiotics11121688

**Published:** 2022-11-23

**Authors:** Saharuetai Jeamsripong, Varangkana Thaotumpitak, Saran Anuntawirun, Nawaphorn Roongrojmongkhon, Edward R. Atwill, Woranich Hinthong

**Affiliations:** 1Research Unit in Microbial Food Safety and Antimicrobial Resistance, Department of Veterinary Public Health, Faculty of Veterinary Science, Chulalongkorn University, Bangkok 10330, Thailand; 2Department of Veterinary Public Health, Faculty of Veterinary Science, Chulalongkorn University, Bangkok 10330, Thailand; 3Department of Population Health and Reproduction, School of Veterinary Medicine, University of California-Davis, Davis, CA 95616, USA; 4Princess Srisavangavadhana College of Medicine, Chulabhorn Royal Academy, Bangkok 10210, Thailand

**Keywords:** antimicrobial resistance, coastal aquaculture, *E. coli*, ESBL, resistance genes, *Salmonella*, *V. cholerae*, *V. parahaemolyticus*

## Abstract

The occurrence of waterborne antimicrobial-resistant (AMR) bacteria in areas of high-density oyster cultivation is an ongoing environmental and public health threat given the popularity of shellfish consumption, water-related human recreation throughout coastal Thailand, and the geographical expansion of Thailand’s shellfish industry. This study characterized the association of phenotypic and genotypic AMR, including extended-spectrum β-lactamase (ESBL) production, and virulence genes isolated from waterborne *Escherichia coli* (*E. coli*) (*n* = 84), *Salmonella enterica* (*S. enterica*) subsp. *enterica* (*n* = 12), *Vibrio parahaemolyticus* (*V. parahaemolyticus*) (*n* = 249), and *Vibrio cholerae* (*V. cholerae*) (*n* = 39) from Thailand’s coastal aquaculture regions. All *Salmonella* (100.0%) and half of *V. cholerae* (51.3%) isolates harbored their unique virulence gene, *invA* and *ompW*, respectively. The majority of isolates of *V. parahaemolyticus* and *E. coli*, ~25% of *S. enterica* subsp. *enterica*, and ~12% of *V. cholerae*, exhibited phenotypic AMR to multiple antimicrobials, with 8.9% of all coastal water isolates exhibiting multidrug resistance (MDR). Taken together, we recommend that coastal water quality surveillance programs include monitoring for bacterial AMR for food safety and recreational water exposure to water for Thailand’s coastal water resources.

## 1. Introduction

AMR is an important One Health concept involving interconnectedness between humans, animals, and their shared environment, that can pose serious health threats to humans and animals. Every year, more than 700,000 deaths are attributed to AMR infection, and it is estimated that the number of deaths from MDR bacteria will increase to 10 million people by 2050 [1]. The extensive use of antimicrobials in human medicine, veterinary medicine, agriculture, and aquaculture has been implicated in the emergence and dissemination of AMR in the environment. The global consumption of antimicrobials in humans increased more than 60% between 2000 and 2015, especially in low- and middle-income countries, due to simple accessibility and irrational use [2]. The excessive use of antimicrobials directly affects the bacterial community and contributes to the overall selective pressure for AMR in aquatic environments [3]. Moreover, contamination from biocides and heavy metals in aquatic environments can also contribute to AMR selection due to their cross-resistance [4].

Studies of AMR in humans and animals have been widely investigated in contrast to environmental monitoring and surveillance of AMR. Currently, AMR bacteria derived from environmental pollutants are of particular concern since they have become a significant emerging health threat, according to the United Nations Environment Program (UNEP) [5]. AMR and virulence bacteria from anthropogenic activities have been circulated in groundwater and surface water supplied to human communities [6,7]. Water treatment plants are implicated as giant reservoirs of AMR bacteria due to the diversity of bacteria, which includes virulent bacterial strains, AMR, heavy metals, and bioactive ingredients that can promote the likelihood of AMR development [8,9]. The terminal discharge for human wastewater and sewage can often be in coastal areas [10,11]. As a consequence, marine aquatic environments have been identified as critical regional hotspots for investigating how AMR can develop, persist, and disseminate throughout these locations.

Coastal aquaculture is often decentralized and of small scale, but growing consumer demand for cultivated seafood can increase water pollution and enhance the spread of emerging diseases and bacterial contaminants [12]. For example, *E. coli* and *Salmonella* spp. are often found in estuarine water from potential sources such as vertebrate animals, terrestrial runoff from rainfall, or human activities such as sewage or aquaculture, while *Vibrio* spp. are natural inhabitants of marine environments [13,14]. Shiga toxin-producing *E. coli* (STEC) are considered major pathogenic strains that can cause adverse health effects in humans and normally carry *stx*1 and/or *stx*2, which encode for Shiga toxins [15]. STEC infection in humans can cause stomach cramps, diarrhea, vomiting, and may be associated with severe consequences [16]. Pathogenic *Salmonella* spp. carry *invA*, which allow bacteria to invade epithelium cells of the human gastrointestinal tract [17]. Therefore, the *invA* gene has been widely used as a genetic marker to confirm the presence of *Salmonella* isolates [18].

*Vibrio* spp. that can cause human gastrointestinal infection are *V. cholerae* and *V. parahaemolyticus*. Both species produce virulent toxins that cause disease in humans: *V. cholerae* uses *ctxA* to encode cholerae toxin while *V. parahaemolyticus* uses *tdh* and *trh* to encode thermostable direct hemolysin and TDH-related hemolysin [19,20]. *V. cholerae* also harbors specific outer membrane proteins and virulence factors encoded from *ompW*, which can be used as a specific molecular indicator of the species [19]. These virulent bacteria can also harbor AMR genes and become AMR reservoirs that can potentially transfer resistant genes to neighboring bacteria through various horizontal gene transfer mechanisms [21]. Moreover, bacteria that harbor integrons or integrative and conjugative elements can function as effective reservoirs and donors of AMR genes that collectively increase the widespread dissemination of AMR [22,23].

It is estimated that up to 90% of antimicrobial use can be excreted as an active compound or metabolite into the environment, which in turn can foster emergence of AMR and dissemination of resistance determinants in both freshwater and coastal systems [24]. Additionally, the high traditional usage of antimicrobials in Thailand’s aquaculture industry can further select for AMR and disperse associated genetic determinants in estuaries and coastal environments used for shellfish cultivation. Therefore, the objectives of this study were to examine the distribution of bacterial phenotypic and genotypic AMR, their resistance determination, ESBL production, and virulence genes in *E. coli*, *Salmonella* spp., *V. cholerae*, and *V. parahaemolyticus* isolated from coastal water, and to characterize the association between resistance, virulence factors, and ESBL production in these bacterial species from the coastal aquaculture regions.

## 2. Results

### 2.1. Bacterial Confirmation and Identification of Virulence Factors

A total of 384 bacterial isolates including *E. coli* (*n* = 84), *S. enterica* subsp. *enterica* (subspecies I) (*n* = 12), *V. parahaemolyticus* (*n* = 249), and *V. cholerae* (*n* = 39) were isolated from seawater samples from Thailand’s coastal aquaculture regions and subjected to antimicrobial susceptibility testing (AST). All *S. enterica* subsp. *enterica* isolates were positive for *invA*, and 98.4% (*n* = 245/249) of *V. parahaemolyticus* isolates were positive for *tlh*, which is a species-specific gene for *V. parahaemolyticus*. The two virulence factors *tdh* and *trh* were absent from this group of *V. parahaemolyticus* isolates. More than half of the *V. cholerae isolates* (51.3%, *n* = 20) were positive for *ompW*, which is a species-specific gene and virulence factor; in contrast, all 20 isolates were negative for the virulence gene *ctx*, which encodes the cholera toxin. Neither *stx*1 nor *stx*2 were detected in the 84 *E. coli* isolates.

### 2.2. Serotyping of S. enterica *subsp.* enterica and V. cholerae

The distribution of serovars among the twelve seawater isolates of *S. enterica* subsp. *enterica* were Bolton (*n* = 1), Braenderup (*n* = 1), Bruebach (*n* = 1), Chester (*n* = 1), Lamberhurst (*n* = 2), Othmarschen (*n* = 2), Paratyphi B (*n* = 1), Wentworth (*n* = 1), Litchfield (*n* = 1), and Orion (*n* = 1). None of the 20 isolates of *V. cholerae* had O1, O139, or O141.

### 2.3. Phenotypic Resistance

The relative frequency of phenotypic AMR for the 384 bacterial isolates (*n* = 384) was ampicillin (AMP) (52.1%), followed by trimethoprim (TRI) (18.2%), sulfamethoxazole (SUL) (15.9%), streptomycin (STR) (3.4%), and ceftazidime (CAZ) (2.3%) (Table 1). Very few isolates (0.5%) exhibited resistance to ciprofloxacin (CIP), cefotaxime (CTX), and cefpodoxime (CPD), with only 0.3% exhibiting resistance to gentamicin (GEN).

Twenty-five AMR patterns were observed. Among these patterns, AMP (27.6%), AMP-TRI (8.1%), AMP-SUL (5.7%), and TRI (3.4%) were predominant (Table 2). Resistance to a variety of antimicrobial classes was found in *E. coli* and *V. parahaemolyticus* (Figure 1); in contrast, *V. cholerae* (82.1%) and *S. enterica* subsp. *enterica* (75.0%) isolates were mostly susceptible to antimicrobials. MDR was observed in 8.9% of all the bacterial isolates, with *E. coli* (25.0%) and *S. enterica* subsp. *enterica* (16.7%) as the main species conferring the most MDR.

### 2.4. ESBL Production

No ESBL production was detected in any bacterial isolate from the coastal water samples. However, the prevalence of resistance to ceftazidime (2.3%) was higher than for cefotaxime and cefpodoxime (0.5%).

### 2.5. Genotypic Resistance

The most common AMR genes detected in this study were *bla*_TEM_ (5.5%), *tetA* (3.7%), *qnrS* (1.8%), *strA* (1.6%), and *floR* (1.3%) (Table 3), which were predominately found in *E. coli*. The prevalence of mobile genetic elements such as integrase (*int*1) was low, at 1.8% (*n* = 7 of *E. coli* isolates), while no *int*2 and *int*3 were detected (Table 3). Interestingly, four out of seven of the positive *int*1 isolates were MDR. ICEs (*int*_SXT_) were detected in one isolate of *V. cholerae*. None of the bacterial isolates had *mcr*-1, *mcr*-2, and *mcr*-3.

### 2.6. Association between Phenotypic and Genotypic Characterization of Resistance

The association between resistance phenotype, genotype, and resistance determinants was examined for all isolates using Cohen’s kappa coefficient (Table 4). The highest strength of association was observed in four pairs of MDR vs. TET (61%), STR vs. *strA* (0.62), CHL vs. *floR* (72%), and MDR vs. *floR* (72%) (*p* < 0.0001).

Based on multivariate logistic regression, ampicillin-resistant isolates were positively associated with the presence of *bla*_TEM_ (OR = 20.3; *p* < 0.0001). In another analysis, trimethoprim-resistant isolates were associated with the presence of MDR (OR = 5.7; *p* < 0.0001) and *int*1 (OR = 4.7; *p* = 0.015) (Table 5).

## 3. Discussion

Investigation of AMR in the environment is challenging because the bacteria have acquired multiple mechanisms to confer AMR and MDR. This study focused on the distribution of AMR phenotypes and genotypes, and the virulence of *E. coli*, *S. enterica* subsp. *enterica*, *V. parahaemolyticus*, and *V. cholerae* isolated from coastal seawater in dense cultivation areas. Tracking resistant bacteria from coastal seawater is important under a One Health perspective, especially for aquatic products that are usually consumed raw or partially cooked, such as oysters. Therefore, contaminated water with resistant bacteria that is used for coastal aquaculture can pose a serious health risk to human health. Among the 384 bacterial isolates from coastal waters from Thailand’s oyster-producing regions, the most common resistance phenotype was to ampicillin (52.1%), which is consistent with previous studies that also found a high prevalence of ampicillin resistance in coastal water [25,26,27].

The overall prevalence of MDR in this study was less than 10%, with the majority of the MDR bacteria being *E. coli* (25.0%) and *S. enterica* subsp. *enterica* (16.7%). In this study, *E. coli* also had the highest diversity of resistance patterns compared to other bacterial species, indicating that different bacterial species from the same coastal environment can acquire and/or maintain different resistance traits. This finding indicates that *E. coli* is potentially a reservoir and/or mode of introduction of drug resistance for coastal environments used for shellfish production in Thailand. A recent study indicated that the sources of bacterial contamination that causes marine pollution were anthropogenic activities, aquaculture, agriculture, industry, etc. [28]. In low- and middle-income countries, untreated water from inadequate and ineffective facilities has been shown to be a significant source of bacterial contamination [29]. Thermotolerant coliforms are the main bacteria in fecal coliforms that are usually present in the intestinal tracts of human and warm-blooded animals, which can indicate fecal contamination in humans, animals, water, food, and the environment [30,31]. In Thailand, the Department of Pollution Control sets the microbiological standards of coastal seawater (total coliforms, thermotolerant coliforms, and *Enterococci*), but does not regulate for the presence of resistant bacteria. Therefore, guidelines may be needed to reduce AMR bacterial contamination in coastal environments, especially for regions used for shellfish production and human recreation.

Among the 384 isolates, all *S. enterica* subsp. *enterica* isolates (*n* = 12) harbored *invA*. A diversity of *S. enterica* subsp. *enterica* serovars was observed in this study, indicating that multiple sources of contamination may exist for this bacterial pathogen. For example, *S. enterica* subsp. *enterica* serovars Lamberhurst and Othmarschen were reported in poultry and humans [32,33]. *S. enterica* subsp. *enterica* serovars Braenderup, Bruebach, Chester, Paratyphi B, Wentworth, Litchfield, and Orion have been isolated from shell eggs, papaya, wastewater, companion animals, livestock animals, and humans [34,35,36,37,38]. The *stx*1 and *stx*2 genes are bacterial toxins found in various serogroups of *E. coli*, but neither *stx*1 nor *stx*2 was detected in this study. Given our sample size of 84 *E. coli*, perhaps it is not surprising that *stx* genes are not found in this bank of *E. coli,* given that it has been estimated that one in a 1000 fecal coliform isolates harbors the *stx* gene [39,40].

The *tdh* (thermostable direct hemolysin) and *trh* (TDH-related hemolysin) are major virulence factors of *V. parahaemolyticus*. In this study, both genes were not present in our coastal water isolates. However, other virulence indicators, including type 3 secretion systems T3SS1 and T3SS2β found in pathogenic *V. parahaemolyticus*, have been reported in seafood samples [41]. For *V. cholerae*, more than half (51.3%) of the isolates were positive for *ompW*, which codes for an outer membrane protein, but none harbored the *ctx* gene which codes for the cholera toxin. This result agrees with previous studies showing that isolates of environmental *V. cholerae* are often not positive for the *ompW* gene [42,43]. The *V. cholerae* serogroups O1, O139, and O141 were also not found in this study. The distribution of serogroup O1 has been reported in environmental samples in northern Thailand, which contrasts with the results of this study [44]. However, non O1/O139 isolated from coastal area of southern Thailand was reported in humans with gastroenteritis [45].

ESBL-producing bacteria have been implicated in impacts on public health due to limited therapeutic options following human infections. The rapid spread of ESBL-producing bacteria in coastal environments increases concerns because of the widespread occurrence of gram-negative bacteria. In this study, ESBL producers were not detected in any of our coastal bacterial isolates. However, the cefotaxime-hydrolysing β-lactamase isolated in Munich (CTX-M) families, including CTX-M-15, CTX-M-14, and CTX-M-27, have been commonly found worldwide [46]. Previous studies reported ESBL producers in lagoon, recreational water, and wastewater [47,48]. Moreover, the spread of carbapenem-resistant *Enterobacterales* has been an emerging threat in coastal and estuarine water [49]. The *bla*_TEM_ (5.5%) was the main resistance gene detected in this study, which agrees with a previous study [50]. Inversely, a previous study of *E. coli* isolated from coastal water contained a variety of *bla*_CTX-M_ genes [51]. Other genes, including *tetA* (3.7%), *qnrS* (1.8%), *strA* (1.6%), and *floR* (1.3%), were detected at low prevalence this study. On the other hand, *sul*1 and *sul*2 were the most abundant AMR genes in the coastal mariculture system in China [52]. The absence of *mcr*-1, *mcr*-2, and *mcr*-3 was observed in this study, which is in contrast to a recent study in Brazil which found *mcr*-1 in *E. coli* isolated from coastal waters [53]. These differences in the geographical distribution of AMR contamination for coastal environments suggest that the processes of AMR contamination and persistence can vary widely from region to region.

Novel resistance mechanisms are associated with mobile genetic elements, which can facilitate the widespread dissemination of resistance determinants in the environment [54]. In this study, integrase (*int*1) was observed at a low level (1.8%), limited mostly to isolates of MDR *E. coli*. This observation raises concern of transferable MDR genes between intra- and inter-bacterial species, because integrons are located in transferable plasmids and conserved DNA sequences carrying gene cassettes with resistance genes, which can facilitate the spread of multiple resistant genes simultaneously [55]. In this study, one isolate of *V. cholerae* had *int*_SXT_. As a consequence, it may be prudent for coastal water quality monitoring programs to also include AMR surveillance for these mobile genetic elements.

Cohen’s kappa analyses are generally used to test the agreement between two test methods. This study found strong associations (kappa agreement: 0.61–0.80) between MDR vs. TET, STR vs. *strA*, CHL vs. *floR*, and MDR vs. *floR*, each with statistical significance. For example, the MDR isolates were associated with the presence of resistance to tetracycline and *floR*, while streptomycin- and chloramphenicol-resistant bacteria were associated with their corresponding genes (*strA* and *floR*). Numerous other pairs of association were also observed, indicating relatively common association between resistant phenotypes, genotypes, and their determinants. Regarding the inferences from the logistic regression analyses, ampicillin-resistant isolates exhibited 20-times higher odds of carrying *bla*_TEM_ (OR = 20.3) compared to ampicillin-susceptible isolates (*p* < 0.0001). This finding indicates that ampicillin-resistant isolates collected from Thailand’s coastal aquaculture regions may preferentially harbor *bla*_TEM_. A previous study recommended that *bla*_TEM_ might be a good indicator for AMR resistance genes in wastewater [56]. Additional regression analyses found that the trimethoprim-resistant isolates were positively associated with MDR (OR = 5.7; *p* < 0.0001) and *int*1 (OR = 4.7; *p* = 0.015). This observation suggests that mobile genetic elements may play a significant role in MDR bacterial development. Further studies using whole genome sequencing are recommended in order to more fully characterize the genomic basis of bacterial AMR resistance in Thailand’s coastal regions used for shellfish cultivation.

## 4. Materials and Methods

### 4.1. Seawater Sample Collection and Bacterial Isolation

Coastal water samples of 500 mL were collected in sterile bottles from different oyster cultivation areas in Thailand, including Surat Thani (9°12′737″ N, 99°27′276″ E), Chanthaburi (8°22′587″ N, 98°35′846″ E), Trat (12°04′610″ N, 102°35′843″ E), Phetchaburi (13°15′843″ N, 99°59′299″ E), Chonburi (13°20′482″ N, 100°55′054″ E), and Phang Nga (8°22′587″ N, 98°35′846″ E) provinces, during February 2021 to January 2022.

Bacterial confirmation followed standard methods. *E. coli* was determined using Levine’s Eosin Methylene Blue (L-EMB) agar (Difco, Becton, Dickinson and Company, Sparks, MD, USA). The colonies showing flat, dark center, with or without metallic sheen, were collected and confirmed by biochemical tests, including triple sugar iron (TSI) agar and indole test [57]. Xylose Lysine Deoxycholate (XLD) (Difco) and MacConkey Agar (Difco) were used for *Salmonella* determination. Presumptive colonies showing pink color with or without black center were confirmed biochemically as *Salmonella* using TSI and citrate test [58]. CHROMagar™ *Vibrio* Agar (HiMedia Laboratories Ltd., Mumbai, India) was used for *Vibrio* spp. determination with colonies showing mauve and green-blue color identified as *V. parahaemolyticus* and *V. cholerae*, respectively. Oxidase test and arginine glucose slants were used to confirm isolates [59]. The positive control strains were *E. coli* ATCC™ 25,922, *S. enterica* serovar Typhimurium ATCC™ 14,028, *V. parahaemolyticus* ATCC™ 17,802, and *V. cholerae* non-O1/non-O139 ATCC™ 14,733, respectively.

One confirmed bacterial isolate of *E. coli*, *V. parahaemolyticus*, and *V. cholerae* per one positive sample, and up to five *Salmonella* isolates per one positive sample were sub-cultured and stored in 20% glycerol at −80 °C in the Department of Veterinary Public health, Faculty of Veterinary Science from Chulalongkorn University.

### 4.2. Serotyping of S. enterica *subsp.* enterica and V. cholerae

All *Salmonella* isolates were serotyped by detection of somatic (O) and flagella (H) antigens using slide agglutination, according to the Kauffmann–White scheme [60] with available commercial antiserum (S&A Reagents Lab, Bangkok, Thailand).

All *V. cholerae* isolates were serotyped using a slide agglutination test with polyvalent *V. cholerae* O1, monoclonal *V. cholerae* O139, and monoclonal *V. cholerae* O141 antiserum (S&A Reagents Lab, Bangkok, Thailand).

### 4.3. Antimicrobial Susceptibility Testing

AST was performed on all isolates using a standard agar dilution method [61]. Eight antimicrobial drugs, including ampicillin, chloramphenicol, ciprofloxacin, gentamicin, streptomycin, sulfamethoxazole, tetracycline, and trimethoprim, were selected based on their importance in human and veterinary medicine. Clinical breakpoints and epidemiological cut-off values of *E. coli*, *S. enterica* subsp. *enterica*, *V. parahaemolyticus*, and *V. cholerae* were based on available standard protocols. The antimicrobials with their clinical breakpoints for *E. coli* and *S. enterica* subsp. enterica (in parentheses) are ampicillin (32 µg/mL), chloramphenicol (32 µg/mL), ciprofloxacin (>1 µg/mL), gentamicin (>8 µg/mL), streptomycin (32 µg/mL), sulfamethoxazole (512 µg/mL), tetracycline (16 µg/mL), and trimethoprim (16 µg/mL) [61]. For *V. parahaemolyticus* and *V. cholerae*, the MIC breakpoints are ampicillin (32 µg/mL), chloramphenicol (32 µg/mL), ciprofloxacin (4 µg/mL), gentamicin (>8 µg/mL), streptomycin (64 µg/mL), sulfamethoxazole (76 µg/mL), tetracycline (16 µg/mL), and trimethoprim (4 µg/mL) [62]. *Staphylococcus aureus* ATCC 29213, *E. coli* ATCC 25922, and *Pseudomonas aeruginosa* ATCC 27853 were used as positive control strains. Resistance to at least three groups of antimicrobials was considered MDR.

### 4.4. Determination of ESBL Production

The disc diffusion method was performed on all bacterial isolates according to the Clinical and Laboratory Standards Institute (CLSI) guidelines [61]. The susceptibility to ceftazidime (30 µg), cefotaxime (30 µg), and cefpodoxime (10 µg) (Oxoid, Basingstoke, UK) was used for the screening test. Resistance to at least one of these cephalosporins was then confirmed using a combination disk diffusion method of ceftazidime (30 µg), cefotaxime (30 µg), and these two disks combined with clavulanic acid. A difference in the inhibition zone between single cephalosporin and cephalosporins containing clavulanic acid greater than 5 mm was classified as a positive ESBL-producing isolate.

### 4.5. Genotypic Characterization of AMR and Virulence Genes by Polymerase Chain Reaction (PCR)

The DNA template was prepared using the whole cell boiling method [63]. Briefly, the bacterial isolate was streaked onto nutrient agar (Difco) and incubated overnight at 37 °C. An individual colony was transferred to an Eppendorf tube containing 150 µL of rNase-free water. The suspension was mixed, heated, and immediately placed on ice, after which the suspension was centrifuged at 11,000 rpm for 5 min. DNA-containing supernatant was collected for PCR template.

All primers for resistance genes, their determinants, and virulence-encoded genes are listed in Table 6 (21 antimicrobials, 4 resistance determinants, 7 virulence genes). Resistance genes were selected to correspond with AMR phenotypes, β-lactam and ESBL production (*bla*_TEM_, *bla*_SHV_, *bla*_CTX-M_, and *bla*_PSE_); carbapenem (*bla*_NDM_ and *bla*_OXA_); phenicols (*floR* and *cmlA*); erythromycin (*ermB*); quinolones (*qnrS*); gentamicin (*aadA*1); tetracyclines (*tetA* and *tetB*); streptomycin (*strA*); sulfonamide (*sul*1 and *sul*2); trimethoprim (*dfrA*1 an

d *dfrA*12); and colistin (*mcr*-1, *mcr*-2, and *mcr*-3). AMR determinants such as integrons (*int*1, *int*2, and *int*3) and the SXT element (*int*_SXT_) were also screened for. The presence of *stx*1 and *stx*2 genes in *E. coli*, *invA* in *S. enterica* subsp. *enterica*, *tdh* and *trh* in *V. parahaemolyticus*, and the outer membrane protein gene (*ompW*) and cholera toxin (*ctx*) *V. cholerae* were also determined. Bacterial isolates, which had been confirmed by PCR and sequencing of relevant genes and elements in previous studies, were used as positive control strains [14,64,65,66,67].

**Table 6 antibiotics-11-01688-t006:** Primers used in this study.

Gene	Primer	Oligonucleotide Sequences (5′-3′)	Product Size (bp)	Reference
Genotype				
*bla* _TEM_	*bla*_TEM_-F	GCGGAACCCCTATTT	964	[68]
	*bla*_TEM_-R	TCTAAAGTATATATGAGTAAACTTGGTCTGAC		
*bla* _SHV_	*bla*_SHV_-F	TTCGCCTGTGTATTATCTCCCTG	854	[69]
	*bla*_SHV_-R	TTAGCGTTGCCAGTGYTG		
*bla* _CTX-M_	*bla*_CTX-M_-F	CGATGTGCAGTACCAGTAA	585	[70]
	*bla*_CTX-M_-R	AGTGACCAGAATCAGCGG		
*bla* _PSE_	*Bla*_PSE_-F	GCTCGTATAGGTGTTTCCGTTT	575	[71]
	*bla*_PSE_-R	CGATCCGCAATGTTCCATCC		
*bla* _NDM_	*bla*_NDM_-F	GGTTTGGCGATCTGGTTTTC	621	[72]
	*bla*_NDM_-R	CGGAATGGCTCATCACGATC		
*bla* _OXA_	*bla*_OXA_-F	ACACAATACATATCAACTTCGC	813	[73]
	*bla*_OXA_-R	AGTGTGTGTTTAGAATGGTGATC		
*floR*	*floR*-F	ATGGTGATGCTCGGCGTGGGCCA	800	[74]
	*floR*-R	GCGCCGTTGGCGGTAACAGACACCGTGA		
*cmlA*	*cmlA*-F	TGGACCGCTATCGGACCG	641	[64]
	*cmlA*-R	CGCAAGACACTTGGGCTGC		
*ermB*	*ermB*-F	AGACACCTCGTCTAACCTTCGCTC	640	[75]
	*ermB*-R	TCCATGTACTACCATGCCACAGG		
*qnrS*	*qnrS*-F	GCAAGTTCATTGAACAGGGT	428	[76]
	*qnrS*-R	TCTAAACCGTCGAGTTCGGCG		
*addA*1	*addA*1-F	CTCCGCAGTGGATGGCGG	631	[64]
	*addA*1-R	GATCTGCGCGCGAGGCCA		
*tetA*	*tetA*-F	GCTGTCGGATCGTTTCGG	658	[64]
	*tetA*-R	CATTCCGAGCATGAGTGCC		
*tetB*	*tetB*-F	CTGTCGCGGCATCGGTCAT	615	[64]
	*tetB*-R	CAGGTAAAGCGATCCCACC		
*strA*	*strA*-F	TGGCAGGAGGAACAGGAGG	405	[64]
	*strA*-R	AGGTCGATCAGACCCGTGC		
*sul*1	*sul*1-F	CGGCGTGGGCTACCTGAACG	433	[77]
	*sul*1-R	GCCGATCGCGTGAAGTTCCG		
*sul*2	*sul*2-F	CGGCATCGTCAACATAACCT	721	[77]
	*sul*2-R	TGTGCGGATGAAGTCAGCTC		
*dfrA*1	*dfrA*1-F	GGAGTGCCAAAGGTGAACAGC	367	[78]
	*dfrA*1-R	GAGGCGAAGTCTTGGGTAAAAAC		
*dfrA*12	*dfrA*12-F	TTCGCAGACTCACTGAGGG	330	[79]
	*dfrA*12-R	CGGTTGAGACAAGCTCGAAT		
*mcr*-1	*mcr*-1-F	AGTCCGTTTGTTCTTGTGGC	320	[79]
	*mcr*-1-R	AGATCCTTGGTCTCGGCTTG		
*mcr*-2	*mcr*-2-F	CAAGTGTGTTGGTCGCAGTT	715	[79]
	*mcr*-2-R	TCTAGCCCGACAAGCATACC		
*mcr*-3	*mcr*-3-F	AAATAAAAATTGTTCCGCTTATG	929	[79]
	*mcr*-3-R	AATGGAGATCCCCGTTTTT		
Integrons				
*int*1	*int*1-F	CCTGCACGGTTCGAATG	497	[80]
	*Int*1-R	TCGTTTGTTCGCCCAGC		
*int*2	*int*2-F	GGCAGACAGTTGCAAGACAA	247	[80]
	*int*2-R	AAGCGATTTTCTGCGTGTTT		
*int*3	*int*3-F	CCGGTTCAGTCTTTCCTCAA	155	[80]
	*int*3-R	GAGGCGTGTACTTGCCTCAT		
Integrative and conjugative elements				
*int* _sxt_	*int*_SXT_-F	GCTGGATAGGTTAAGGGCGG	592	[80]
	*int*_SXT_-R	CTCTATGGGCACTGTCCACATTG		
Virulence genes of *E. coli*				
*stx*1	*stx*-1-F	CAACACTGGATGATCTCAG	349	[81]
	*stx*-1-R	CCCCCTCAACTGCTAATA		
*stx*2	*stx*-2-F	ATCAGTCGTCACTCACTGGT	110	[81]
	*stx*-2-R	CTGCTGTCACAGTGACAAA		
Species-specific and virulence genes of *S. enterica subsp. enterica*				
*invA*	*invA*-F	GTGAAATTATCGCCACGTTCGGGCAA	284	[82]
	*invA*-R	TCATCGCACCGTCAAAGGAACC		
Species-specific * and virulence genes of *V. parahaemolyticus*				
*tlh* *	*tlh*-F	AAAGCGGATTATGCAGAAGCACTG	450	[83]
	*tlh*-R	GCTACTTTCTAGCATTTTCTCTGC		
*tdh*	*tdh*-F	GTAAAGGTCTCTGACTTTTGGAC	269	[83]
	*tdh*-R	TGGAATAGAACCTTCATCTTCACC		
*trh*	*trh*-F	TTGGCTTCGATATTTTCAGTATCT	500	[83]
	*trh*-R	CATAACAAACATATGCCCATTTCCG		
Species-specific * and virulence genes of *V. cholerae*				
*ompW* *	*ompW*-F	CACCAAGAAGGTGACTTTATTGTG	588	[42]
	*ompW*-R	GAACTTATAACCACCCGCG		
*ctx*	*ctx*-F	CAGTCAGGTGGTCTTATGCCAAGAGG	167	[84]
	*ctx*-R	CCCACTAAGTGGGCACTTCTCAAACT		

* Indicted specie-specific genes for *V. parahaemolyticus* and *V. cholerae*.

PCR was conducted following the manufacturer’s instructions. A 5 µL of DNA template, 25 µL of TopTaq Master Mix (Qiagen^®^, Stockach, Germany), 5 µL of coralLoad, 2 µL of each forward and reverse primer, and 11 µL of sterile rNase free water were utilized. The PCR amplification was performed based on Tpersonal combi model (Biometra^®^, Göttingen, Germany). PCR products were then separated on 1.5% (*w*/*v*) agarose gel, stained with Redsafe^TM^ nucleic acid staining solution (Intron Biotechnology, Seongnam, Republic of Korea), and photographed using Omega Fluor™ gel documentation system (Aplegen, CA, USA).

### 4.6. Statistical Analyses

Descriptive statistics were used to characterize the prevalence of resistance, AMR distribution, ESBL production, virulence genes, integrons, and SXT element in *E. coli*, *Salmonella*, *V. parahaemolyticus*, and *V. cholerae*. Cohen’s kappa coefficient was used to determine the agreement between pairs of phenotypes and genotypes of AMR and resistance determinants for all isolates. The interpretation of the kappa coefficient, expressed as a strength of agreement, was: <0.00: poor; 0.00–0.20: slight; 0.21–0.40: fair; 0.41–0.60: moderate; 0.61–0.80: substantial; 0.81–1.00: almost perfect [85].

Multivariate logistic regression analysis was performed to characterize the association between the most common resistant phenotypes, AMR determinants, virulence genes, and ESBL production. Odds ratios (OR) were used to identify the magnitude of the observed association. The interpretation of ORs was indicated as OR > 1: positive association; OR < 1: negative association; OR = 1: no association. Two-sided hypothesis testing together with likelihood ratio test were used with a *p* ≤ 0.05 to decide statistical significance. All statistical analyses were performed with Stata version 14.0 (StataCorp, College Station, TX, USA).

## 5. Conclusions

The occurrence of waterborne AMR bacteria in areas of high-density oyster cultivation is an ongoing environmental and public health threat consistent with the One Health concept, especially given the popularity of shellfish consumption, water-related human recreation throughout coastal Thailand, and geographical expansion of the shellfish industry. Waterborne isolates of *E. coli*, *S. enterica* subsp. *enterica*, *V. parahaemolyticus*, and, to a lesser extent, *V. cholerae* from coastal Thailand exhibited phenotypic AMR for a wide variety of antimicrobials and for *E. coli* possessing concurrent genomic AMR. Although coastal seawater is regulated for excessive total coliforms, fecal coliforms, and *Enterococci*, current water quality monitoring does not include bacteria surveillance for phenotypic and/or genotypic AMR. Therefore, in addition to tracking and preventing key sources of bacterial contamination and promoting proper treatment of wastewater before release to Thailand’s coastal water resources, we recommend that water quality surveillance programs also include monitoring excessive levels of bacterial AMR to better protect shellfish food safety, water-related body contact recreation, and to reduce AMR contamination for Thailand’s coastal water resources.

## Figures and Tables

**Figure 1 antibiotics-11-01688-f001:**
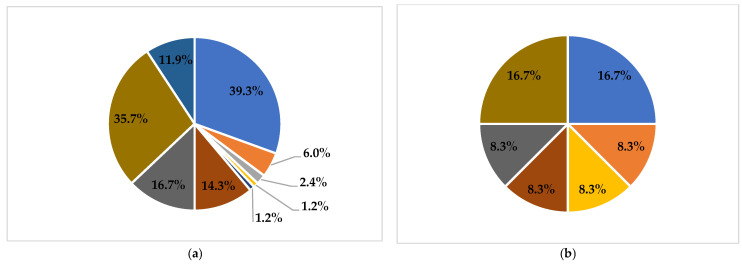
Proportion of phenotypic AMR for different species of bacteria. (**a**): *E. coli*; (**b**): *S. enterica* subsp. *enterica*; (**c**): *V. parahaemolyticus*; (**d**): *V. cholerae*.

**Table 1 antibiotics-11-01688-t001:** Phenotypic characterization of AMR in *E. coli*, *Salmonella*, *V. parahaemolyticus*, and *V. cholerae* isolated from seawater from coastal Thailand.

Antimicrobial Agent	Prevalence of Phenotypic AMR (%)				
	*E. coli* (*n* = 84)	*S. enterica* subsp. *enterica* (*n* = 12)	*V. parahaemolyticus* (*n* = 249)	*V. cholerae* (*n* = 39)	Total (*n* = 384)
Ampicillin	33 (39.3)	2 (16.7)	162 (65.1)	3 (7.7)	200 (52.1)
Chloramphenicol	5 (6.0)	1 (8.3)	0 (0)	0 (0)	6 (1.6)
Ciprofloxacin	2 (2.4)	0 (0)	0 (0)	0 (0)	2 (0.5)
Ceftazidime	1 (1.2)	1 (8.3)	5 (2.0)	2 (5.1)	9 (2.3)
Cefotaxime	0 (0)	0 (0)	2 (0.8)	0 (0)	2 (0.5)
Cefpodoxime	0 (0)	0 (0)	2 (0.8)	0 (0)	2 (0.5)
Gentamicin	1 (1.2)	0 (0)	0 (0)	0 (0)	1 (0.3)
Streptomycin	12 (14.3)	1 (8.3)	0 (0)	0 (0)	13 (3.4)
Sulfamethoxazole	14 (16.7)	1 (8.3)	42 (16.9)	4 (10.3)	61 (15.9)
Tetracycline	30 (35.7)	2 (16.7)	1 (0.4)	0 (0)	33 (8.6)
Trimethoprim	10 (11.9)	0 (0)	60 (24.1)	0 (0)	70 (18.2)
MDR ^1^	21 (25.0)	2 (16.7)	11 (4.4)	0 (0)	34 (8.9)

^1^ MDR: multidrug resistance.

**Table 2 antibiotics-11-01688-t002:** Phenotypic resistance patterns of *E. coli*, *S. enterica* subsp. *enterica*, *V. parahaemolyticus*, and *V. cholerae* isolated from seawater from coastal Thailand.

AMR Pattern	No. (%)				
	*E. coli* (*n* = 84)	*S. enterica* subsp. *enterica* (*n* = 12)	*V. parahaemolyticus* (*n* = 249)	*V. cholerae* (*n* = 39)	Total (*n* = 384)
Susceptible	47 (56.0)	9 (75.0)	59 (23.7)	32 (82.1)	147 (38.3)
AMP	6 (7.1)	0 (0)	99 (39.8)	1 (2.6)	106 (27.6)
AMP-CAZ	0 (0)	0 (0)	1 (0.4)	0 (0)	1 (0.3)
AMP-CAZ-CPD-CTX	0 (0)	0 (0)	2 (0.8)	0 (0)	2 (0.5)
AMP-CAZ-SUL-TRI	0 (0)	0 (0)	1 (0.4)	0 (0)	1 (0.3)
AMP-CAZ-TET	1 (1.2)	0 (0)	0 (0)	0 (0)	1 (0.3)
AMP-CHL-CIP-SUL-TET	1 (1.2)	0 (0)	0 (0)	0 (0)	1 (0.3)
AMP-CHL-CIP-SUL-TET-TRI	1 (1.2)	0 (0)	0 (0)	0 (0)	1 (0.3)
AMP-CHL-SUL-TET	2 (2.4)	1 (8.3)	0 (0)	0 (0)	3 (0.8)
AMP-CHL-SUL-TET-TRI	1 (1.2)	0 (0)	0 (0)	0 (0)	1 (0.3)
AMP-GEN-SUL-TET-TRI	1 (1.2)	0 (0)	0 (0)	0 (0)	1 (0.3)
AMP-STR-SUL	1 (1.2)	0 (0)	0 (0)	0 (0)	1 (0.3)
AMP-STR-SUL-TET	1 (1.2)	0 (0)	0 (0)	0 (0)	1 (0.3)
AMP-STR-SUL-TET-TRI	4 (4.8)	0 (0)	0 (0)	0 (0)	4 (1.0)
AMP-STR-TET	6 (7.1)	1 (8.3)	0 (0)	0 (0)	7 (1.8)
AMP-SUL	0 (0)	0 (0)	20 (8.0)	2 (5.1)	22 (5.7)
AMP-SUL-TET-TRI	2 (2.4)	0 (0)	0 (0)	0 (0)	2 (0.5)
AMP-SUL-TRI	0 (0)	0 (0)	8 (3.2)	0 (0)	8 (2.1)
AMP-TET	6 (7.1)	0 (0)	0 (0)	0 (0)	6 (1.6)
AMP-TRI	0 (0)	0 (0)	31 (12.4)	0 (0)	31 (8.1)
CAZ	0 (0)	1 (8.3)	1 (0.4)	2 (5.1)	4 (1.0)
SUL	0 (0)	0 (0)	7 (2.8)	2 (5.1)	9 (2.3)
SUL-TRI	0 (0)	0 (0)	6 (2.4)	0 (0)	6 (1.6)
TET	3 (3.6)	0 (0)	0 (0)	0 (0)	3 (0.8)
TET-TRI	1 (1.2)	0 (0)	1 (0.4)	0 (0)	2 (0.5)
TRI	0 (0)	0 (0)	13 (5.2)	0 (0)	13 (3.4)

AMP: ampicillin; CAZ: ceftazidime; CHL: chloramphenicol; CIP: ciprofloxacin; CPD: cefpodoxime; CTX: cefotaxime; GEN: gentamicin; STR: streptomycin; SUL: sulfamethoxazole; TET: tetracycline; TRI: trimethoprim.

**Table 3 antibiotics-11-01688-t003:** Genotypic characterization of AMR and their determinants.

Gene	Prevalence of Genotypic AMR (%)				
	*E. coli* (*n* = 84)	*Salmonella* spp. (*n* = 12)	*V. parahaemolyticus* (*n* = 249)	*V. cholerae* (*n* = 39)	Total (*n* = 384)
Genotype					
*bla* _TEM_	21 (25.0)	0 (0)	0 (0)	0 (0)	21 (5.5)
*bla* _SHV_	0 (0)	0 (0)	0 (0)	0 (0)	0 (0)
*bla* _CTX-M_	0 (0)	0 (0)	0 (0)	0 (0)	0 (0)
*bla* _PSE_	0 (0)	0 (0)	0 (0)	0 (0)	0 (0)
*bla* _NDM_	0 (0)	0 (0)	0 (0)	0 (0)	0 (0)
*bla* _OXA_	0 (0)	0 (0)	0 (0)	0 (0)	0 (0)
*floR*	4 (4.8)	1 (8.3)	0 (0)	0 (0)	5 (1.3)
*cmlA*	2 (2.4)	1 (8.3)	0 (0)	0 (0)	3 (0.8)
*ermB*	0 (0)	0 (0)	0 (0)	0 (0)	0 (0)
*qnrS*	7 (8.3)	0 (0)	0 (0)	0 (0)	7 (1.8)
*aadA*1	0 (0)	0 (0)	0 (0)	0 (0)	0 (0)
*tetA*	14 (16.7)	0 (0)	0 (0)	0 (0)	14 (3.7)
*tetB*	2 (2.4)	0 (0)	0 (0)	0 (0)	2 (0.5)
*strA*	6 (7.1)	0 (0)	0 (0)	0 (0)	6 (1.6)
*sul*1	1 (1.2)	0 (0)	0 (0)	0 (0)	1 (0.3)
*sul*2	1 (1.2)	0 (0)	0 (0)	0 (0)	1 (0.3)
*dfrA*1	0 (0)	0 (0)	0 (0)	0 (0)	0 (0)
*dfrA*12	3 (3.6)	0 (0)	0 (0)	0 (0)	3 (0.8)
*mcr*-1	0 (0)	0 (0)	0 (0)	0 (0)	0 (0)
*mcr*-2	0 (0)	0 (0)	0 (0)	0 (0)	0 (0)
*mcr*-3	0 (0)	0 (0)	0 (0)	0 (0)	0 (0)
Integrons					
*int*1	7 (8.3)	0 (0)	0 (0)	0 (0)	7 (1.8)
*int*2	0 (0)	0 (0)	0 (0)	0 (0)	0 (0)
*int*3	0 (0)	0 (0)	0 (0)	0 (0)	0 (0)
Integrative and conjugative elements					
*int* _SXT_	0 (0)	0 (0)	0 (0)	1 (2.6)	1 (0.3)

**Table 4 antibiotics-11-01688-t004:** Kappa analysis among phenotype, genotype, and resistance determinants (*n* = 384).

Resistance		% Agreement		Kappa	Std. Err. ^1^	Degree of Agreement	*p*-Value
		Observed	Expected				
Phenotype	Phenotype						
MDR	AMP	56.7	48.4	0.16	0.03	Slight	<0.0001
MDR	CAZ	91.1	89.4	0.16	0.04	Slight	0.0001
MDR	CHL	93.0	90.1	0.29	0.04	Fair	<0.0001
MDR	CIP	91.9	91.0	0.11	0.02	Slight	<0.0001
MDR	CPD	91.2	91.0	0.11	0.02	Slight	<0.0001
MDR	CTX	91.9	91.0	0.11	0.02	Slight	<0.0001
MDR	GEN	91.6	91.2	0.05	0.02	Slight	<0.0001
MDR	STR	94.5	88.8	0.51	0.04	Moderate	<0.0001
MDR	SUL	88.0	78.2	0.45	0.05	Moderate	<0.0001
MDR	TET	94.0	84.5	0.61	0.05	Substantial	<0.0001
MDR	TRI	82.5	76.3	0.26	0.05	Fair	<0.0001
Phenotype	Genotype						
AMP	*bla* _TEM_	52.9	48.1	0.09	0.02	Slight	<0.0001
AMP	*int*1	49.2	48.0	0.02	0.01	Slight	0.036
CAZ	*int*1	96.4	95.9	0.11	0.05	Slight	0.018
CHL	*floR*	99.2	97.2	0.72	0.05	Substantial	<0.0001
CHL	*cmlA*	98.7	97.7	0.44	0.05	Moderate	<0.0001
CHL	*int*1	97.1	96.7	0.14	0.05	Slight	0.003
TET	*tetA*	94.5	88.4	0.53	0.05	Moderate	<0.0001
TET	*tetB*	91.9	91.0	0.11	0.02	Slight	<0.0001
STR	*strA*	98.2	95.2	0.62	0.05	Substantial	0.0001
STR	*int*1	95.8	94.9	0.18	0.05	Slight	<0.0001
SUL	*sul*1	84.4	83.9	0.03	0.01	Slight	<0.05
SUL	*int*1	84.4	82.9	0.09	0.03	Slight	0.0013
TET	*int*1	93.2	89.9	0.33	0.04	Fair	<0.0001
TRI	*dfrA*12	82.0	81.3	0.04	0.02	Slight	<0.05
TRI	*int*1	82.6	80.6	0.10	0.03	Slight	0.0001
MDR	*bla* _TEM_	91.9	87.1	0.37	0.05	Fair	<0.0001
MDR	*cmlA*	91.6	90.7	0.10	0.03	Slight	0.0002
MDR	*dfrA*12	91.6	90.7	0.10	0.03	Slight	0.0002
MDR	*floR*	99.2	97.2	0.72	0.05	Substantial	<0.0001
MDR	*qnrS*	91.6	90.0	0.18	0.04	Slight	<0.0001
MDR	*strA*	92.7	90.3	0.25	0.03	Fair	<0.0001
MDR	*tetA*	91.6	88.6	0.27	0.05	Fair	<0.0001
MDR	*tetB*	91.9	91.0	0.11	0.02	Slight	<0.0001
MDR	*sul*1	91.6	91.2	0.05	0.02	Slight	0.0006
MDR	*int*1	92.2	90.0	0.23	0.04	Fair	<0.0001
Genotype	Genotype						
*int*1	*bla* _TEM_	95.8	92.9	0.41	0.04	Moderate	<0.0001
*int*1	*cmlA*	98.4	97.4	0.39	0.05	Fair	<0.0001
*int*1	*dfrA*12	98.4	97.4	0.39	0.05	Fair	<0.0001
*int*1	*floR*	97.9	96.9	0.32	0.05	Fair	<0.0001
*int*1	*qnrS*	97.9	96.4	0.42	0.05	Moderate	<0.0001
*int*1	*strA*	97.7	96.7	0.30	0.05	Fair	<0.0001
*int*1	*sul*2	98.4	98.0	0.25	0.03	Fair	0.0001
*int*1	*tetA*	96.6	94.7	0.37	0.05	Fair	<0.0001
*int*1	*tetB*	98.7	97.7	0.44	0.04	Moderate	<0.0001

^1^ Std. Err.: Standard error; AMP: ampicillin; CAZ: ceftazidime; CHL: chloramphenicol; CIP: ciprofloxacin; CPD: cefpodoxime; CTX: cefotaxime; GEN: gentamicin; MDR: multidrug resistance; STR: streptomycin; SUL: sulfamethoxazole; TET: tetracycline; TRI: trimethoprim.

**Table 5 antibiotics-11-01688-t005:** Multivariate logistic regression analysis of factors associated with ampicillin resistance and MDR in coastal seawater samples stratified by bacterial species (*n* = 384).

Predictor	OR ^1^	Std. Err. ^2^	95% C.I. ^3^	*p*-Value
Model 1				
Ampicillin resistance	Reference group			
*bla* _TEM_	20.333	11.856	6.485–63.758	<0.0001
Intercept	0.984	0.574	0.314–3.084	0.977
AIC ^4^	513.241			
Model 2				
Trimethoprim resistance	Reference group			
MDR ^5^	5.720	1.713	3.180–10.289	<0.0001
*int*1	4.677	2.959	1.354–16.160	0.015
Intercept	0.172	0.088	0.063–0.469	0.001
AIC ^4^	340.641			

^1^ OR: Odds ratio; ^2^ Std. Err.: Standard Error; ^3^ C.I.: Confidence Interval; ^4^ AIC: Akaike information criterion; ^5^ MDR: multidrug resistance.

## Data Availability

Not applicable.

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
