# Peer review of "Molecular Epidemiology of Antimicrobial Resistance and Virulence Profiles of Escherichia coli, Salmonella spp., and Vibrio spp. Isolated from Coastal Seawater for Aquaculture"

_antibiotics, 2022, doi:10.3390/antibiotics11121688_

Round 1
Reviewer 1 Report
Please see the attached file for the Reviewer's Comments and Suggestions for Authors.

Reviewer 3 Report
In line, 34 suggest including the basic objective of the One Health approach, which is to achieve the best health outcomes, recognizing the INTERCONNECTEDNESS between people, animals, plants and their shared environment.
In lines 39 to 40 it is necessary to include that although the generalized AMR has been mainly attributed to the selective pressure generated by the overuse and misuse of antimicrobials, there is recent growing evidence on the selective pressure among bacteria exposed to non-antibiotic compounds used in the agri-food industry, as well as biocides used as disinfectants, antiseptics and preservatives and heavy metals existing in nature and used in agricultural production. This information will answer the origin of the statements contained in lines 49 to 50.
The authors do not use nomenclature rules for Salmonella spp. isolates for isolates in this report. According to the Popoff and Le Minor (2001) classification and nomenclature system two species of this genetically distinct genus are known: Salmonella enterica and Salmonella bongori, and the first is divided into six subspecies (subsp.) Identified by names and digits Romans, which in turn are classified into serotypes or serovares, i.e., antigens sharing groups surface recognized by specific antibodies, while Salmonella bongori has only subspecies bongori (V). This nomenclature is adopted by the international serotyping reference centers for Salmonella spp.
In lines 195 and 198, it is necessary to update the Enterobacteriaceae Family Classification Level for Enterobacterales Order.
In the methodology started in lines 239-251 the use of tests based on biochemical methods, focusing on glycerol utilization, sugar fermentation or the presence of enzymes, for the identification of Enterobacterales, many studies have reported less satisfactory identification. Additionally, biochemical methods may not be reliable for environmental isolates because of small biochemical differences in adequate information on environmental bacteria in databases. Methodologies with greater discriminatory power is required to identify bacterial isolates, for example, Maldi-Tof or microbial identification system, based on comparative sequencing of 16S rDNA.
In line 261 subtitle 4.3. Antimicrobial susceptibility test, the authors use two different standard documents and guidelines for AST methods and interpretation of results. However, they indicate in lines 264 to 266 that listed antimicrobials are used in medicine; it can be concluded that for use in humans. However, the reference number 42 contains interruption points to interpret CIM results and Quality Control Intervals (CQ) for these tests, as well as guidelines on which antimicrobial agents to test and report specific organisms, methods by which they identify resistance mechanisms and how to confirm results in animals.
It is not clear; therefore, what is the reason for the use of standardization used for different water isolates (not animals).
For interpretive Categories Used for Susceptibility Testing of Salmonella spp. and E. coli the reference documents do not have Interpretive Categories and MIC Breakpoints, µg/mL for Trimethoprim and Sulfamethoxazole individually for bacteria of the Order Enterobacterales. In the same way, there is no MIC QC Ranges for Nonfastidious Organisms and Antimicrobial Agents.
It is necessary to replace reference No. 42. I suggest the document CLSI Document M100 Performance Patterns for antimicrobial susceptibility tests, 32nd edition.
Recommendations for testing and reporting the Vibrio spp. (including V. cholerae) are found in the CLSI M45 document is correct.
Round 2
Reviewer 1 Report
Journal: Antibiotics
Manuscript Number: antibiotics-1994528
Article Type: Research Article
Title: Molecular Epidemiology of Antimicrobial Resistance and Virulence Profiles of Bacteria Isolated from Coastal Seawater for Aquaculture
Authors: Saharuetai Jeamsripong, Varangkana Thaotumpitak, Saran Anuntawirun, Nawaphorn Roongrojmongkhon, Edward R. Atwill, and Woranich Hinthong
My comments and suggestions on the manuscript (antibiotics-1994528) have been mostly addressed by the authors, and the manuscript has been revised accordingly.
Author Response
Thank you for your comments and suggestions.
Reviewer 3 Report
In line 50 replace the abbreviation UN with United Nations Environment Program (UNEP).
In line 66 it is necessary to add the term and/or between the types of stx (stx1 and/or stx2). I suggest reading and including the reference https://doi.org/10.2903/j.efsa.2020.5967 .
In lines 66-68, I believe it is important to add the information that the invA gene can be used in the genetic diagnosis of Salmonella species, which would support the use of this virulence marker. I suggest reading and including the reference https://doi.org/10.3390/foods11182924 .
In line 192 replace the term fecal coliforms with "thermotolerant coliforms". I suggest reading and including the references https://doi.org/10.3390/foods11030332 and https://doi.org/10.3390/ijerph18094604 .
In line 222, I suggest replacing the Enterobacterales family with gram-negative bacteria because, although ESBLs have common biochemical properties, the genes that encode these enzymes are of a diverse nature and can be grouped into several families, such as:
1. GES - Guyana-extended spectrum. More prevalent in Pseudomonas aeruginosa than Enterobacterales. Some variants also hydrolyse carbapenems;
2. PER - Pseudomonas extended resistant More prevalent in Pseudomonas aeruginosa and Acinetobacter baumannii than Enterobacterales. I suggest reading and including the reference https://doi.org/10.1093/jacamr/dlab092 .
The results of the MIC (μg/ml) of the isolates and, therefore, of the interpretative categories based on the breakpoints of the reference document are not presented in the article, which prevents the evaluation of all the phenotypic results reported, since the number reference 56 contained in lines 516 and 517 does not exist.
